# In Vitro Assessment of Azole and Amphotericin B Susceptibilities of *Malassezia* spp. Isolated from Healthy and Lesioned Skin

**DOI:** 10.3390/jof8090959

**Published:** 2022-09-13

**Authors:** Wissal Chebil, Najoua Haouas, Elja Eskes, Paul Vandecruys, Sameh Belgacem, Hichem Belhadj Ali, Hamouda Babba, Patrick Van Dijck

**Affiliations:** 1Laboratory of Medical and Molecular Parasitology-Mycology (LR12ES08), Department of Clinical Biology B, Faculty of Pharmacy, University of Monastir, Monastir 5000, Tunisia; 2Laboratory of Molecular Cell Biology, Institute of Botany and Microbiology, Department of Biology, Faculty of Sciences, KU Leuven, Heverlee, 3001 Leuven, Belgium; 3Laboratory of Parasitology-Mycology, Fattouma Bourguiba University Hospital, Monastir 5000, Tunisia; 4Dermatology Department, Fattouma Bourguiba University Hospital, Monastir 5000, Tunisia

**Keywords:** *Malassezia* spp., azole, amphotericin B, antifungal drug resistance, *Pityriasis versicolor*, skin infections, healthy subjects

## Abstract

*Malassezia* yeasts have recently gained medical importance as emerging pathogens associated with a wide range of dermatological and systemic infections. Since standardized methods for in vitro antifungal susceptibility testing have not yet been established for *Malassezia* spp., related diseases are always treated empirically. As a result, a high rate of recurrence and decreased antifungal susceptibility have appeared. Thus, the aims of the study were to assess and analyze the in vitro susceptibility of *Malassezia* isolated from *pityriasis versicolor* (PV) lesions and healthy controls. A total of 58 *Malassezia* strains isolated from PV patients and healthy controls were tested. In vitro antifungal susceptibility testing was conducted using the CLSI broth microdilution with some modifications. *Candida* spp. criteria established in accordance with CLSI guidelines were used for data interpretation. Ketoconazole and posaconazole seemed to be the most effective molecules against *Malassezia* species. However, considerable percentages of itraconazole, fluconazole, and amphotericin B ‘‘resistant’’ strains (27.6%, 29.3%, and 43.1%, respectively) were revealed in this study. *Malassezia furfur*, *M. sympodialis*, and *M. globosa* showed different susceptibility profiles to the drugs tested. These results emphasize the importance of accurately identifying and evaluating the antifungal susceptibility of *Malassezia* species in order to guide a specific and effective treatment regimen.

## 1. Introduction

Lipophilic yeasts of the genus *Malassezia* have been recognized as commensals of the normal skin of humans and warm-blooded animals [1], but species of this genus have become emerging pathogens associated with a wide range of dermatological diseases, such as *Pityriasis versicolor* (PV), Malassezia folliculitis, seborrheic dermatitis/dandruff, atopic dermatitis, and psoriasis [1,2]. PV is the prototypical skin disease etiologically connected to these fungi and is one of the most common superficial mycoses in humans worldwide [3]. This is also the case in Tunisia, with *M. globosa*, *M. sympodialis*, and *M. furfur* the most common species [4,5]. Apart from these superficial infections, *Malassezia* spp. have also been linked to a rising number of severe systemic infections, particularly in immunocompromised patients receiving parenteral nutrition and in premature neonates [6]. More recently, *Malassezia* species have been reported to be associated with gastrointestinal diseases, including inflammatory bowel diseases, gastric and colorectal cancer, as well as with neurodegenerative diseases such as Alzheimer’s and Parkinson’s disease [7,8,9]. 

Despite many attempts to control yeast infections with topical and systemic antifungals, recurrence is often noticed, especially in the case of skin diseases [6]. Moreover, the occurrence of drug resistance phenomena has been suggested by many authors. Indeed, the induction of in vitro fluconazole (FLZ) resistance was described in *M. pachydermatis* [10,11]. In addition, there is clinical evidence of treatment failure with terbinafine in patients with PV, and with ketoconazole (KTZ) in dogs with otitis [12,13,14], and with FLZ or posaconazole (POS) in preventing *M. furfur* fungemia in humans [15,16,17,18]. In addition to the probable acquired resistance, the increased incidence of invasive infections emphasizes the need to study the susceptibility profile of these yeasts in order to choose a specific and accurate treatment [19]. However, due to Malassezia yeasts’ slow growth, lipid dependency, and tendency to form clusters, standardized methods for the in vitro evaluation of antifungal susceptibility are lacking, resulting in variable susceptibility profiles and the lack of clinical breakpoints [20]. Recently, it was suggested that the modified Clinical and Laboratory Standards Institute broth microdilution (CLSI BMD) method using Sabouraud dextrose broth (SDB) added with 1% Tween 80 might be optimal for testing azole susceptibility, mainly for *M. furfur* [20,21]. Moreover, compared with RPMI and Christensen’s urea broth, SDB was the only medium able to identify isolates with high FLZ minimal inhibitory concentrations (MICs) [21]. However, data on the antifungal susceptibility profiles of *Malassezia* spp. coming from healthy and diseased human skin are scarce. Thus, this study aims to (i) assess the in vitro susceptibility of *Malassezia* isolated from PV lesions and healthy controls to FLZ, KTZ, itraconazole (ITZ), POS, and amphotericin B (AMB) using the modified CLSI BMD method, and (ii) analyze the in vitro susceptibility profiles according to clinical origin and species.

## 2. Materials and Methods

### 2.1. Malassezia Strains

In this study, 58 *Malassezia* strains were tested, including *M. furfur* (*n* = 38), *M. sympodialis* (*n* = 11), and *M. globosa* (*n* = 9). *Malassezia* strains were isolated from *Pityriasis versicolor* patients (Group 1: *n* = 40) and from the skin of healthy controls (Group 2: *n* = 18) at the laboratory of Parasitology-Mycology of the university hospital Fattouma Bourguiba (Monastir, Tunisia). All the isolates were pre-identified phenotypically (i.e., macroscopically, microscopically, and physiologically), as previously reported by Guého et al. [22], and molecularly using the PCR multiplex of the Internal Transcribed Spacer ITS region [23]. Briefly, DNA amplification was carried out in a 50 µL reaction volume, with 5 µL of 10x PCR buffer, 100 µmol/L of each of the four dNTPs, 2.5 U of Taq DNA polymerase, and 0.5 mol/L of each oligonucleotide. For all strains, the phenotypic characterization and the PCR analysis was consistent. After the identification step, strains were conserved at −80 °C. Prior to testing, each isolate was sub-cultured at least twice onto modified Dixon agar to ensure purity and viability. 

### 2.2. In Vitro Susceptibility Testing 

The testing of the antifungal susceptibility of *Malassezia* strains was performed using the reference CLSI BMD M27-A3 protocol [24,25] with some modifications. Specifically, Sabouraud dextrose broth (Biolife, Milano, Italy) containing 1% Tween 80 (Sigma Co., Milano, Italy) was used instead of RPMI 1640 medium as previously reported [11,21]. Stock inoculum suspensions were prepared from 4-day-old colonies on modified Dixon agar at 32 °C. Cluster formation was avoided while preparing the inoculum suspension by using sterile glass beads and vortexing. The final concentration of the stock inoculum in sterile distilled water was adjusted to an optical density of 0.4 at 660 nm (2.4 McFarland using a turbidimeter), which is equivalent to 1–5 × 10^6^ colony-forming units (CFU)/mL, as inferred by quantitative plate counts of CFU on Dixon agar. Subsequently, two dilutions (1:100 and 1:20) of the inoculum in the SDB medium were performed, and a total of 100 µL of the final dilution was transferred into a 96-well microtiter plate containing 100 µL of each drug dilution. Duplicates of each strain were incubated for 72 h at 32 °C together with a negative control (medium only).

The following antifungal drugs were supplied by the manufacturers as pure standard compounds: FLZ, KTZ, AMB, POS, and ITZ (all from Sigma-Aldrich, St. Louis, MO, USA). Fluconazole was dissolved in sterile water, whereas the remaining drugs were solubilized in DMSO (1%) (Sigma-Aldrich) and stored at −20 °C until they were used. The concentration of each antifungal drug ranged from 0.016 to 16 mg/L, except for FLZ (i.e., from 0.06 to 64 mg/L). The visual reading of plates was performed after 48 and 72 h of incubation at 32 °C. MICs were defined as the lowest concentration of antifungal drug which visually inhibited 50% of fungal growth for azole antifungals and 100% for AMB as compared with the control wells with no antifungal drug. Quality control strains (*Candida parapsilosis* ATCC 22019 and *Candida krusei* ATCC 6258; American Type Culture Collection, Manassas, VA, USA) were included to assess the accuracy of the drug dilutions and the reproducibility of the results.

### 2.3. Data Interpretation

The data that were obtained are reported as the mean MIC (mMIC), the MIC at which 50% of the isolated were inhibited (MIC50), and the MIC at which 90% of the isolates were inhibited (MIC90). Since breakpoints to classify strains as susceptible (S), susceptible dependent upon dose (SDD), or resistant (R) are lacking for *Malassezia* genus, *Candida* spp. criteria (i.e., KTZ: S ≤ 8 mg/L, R ≥ 16 mg/L; ITZ: S ≤ 0.125 mg/L, SDD 0.25–0.5 mg/L, R ≥ 1 mg/L; FLZ: S ≤ 8 mg/L, SDD 16–32 mg/L, R ≥ 64 mg/L) established in accordance with CLSI guidelines [24,25] were used in the present study. Since breakpoints for AMB and the triazole POS have not yet been established, the breakpoints for voriconazole (S ≤ 1 mg/L, SDD 2 mg/L, R ≥ 4 mg/L) were used for POS, whereas AMB isolates were considered susceptible when MIC was ≤1 mg/L [26].

### 2.4. Statistical Analysis

Data were statistically analyzed using Statistical Package For Social Sciences Software (SPSS for Windows, version 20.0. IBM Corp., Armonk, NY, USA). The Mann–Whitney and Kruskal–Wallis tests were used to evaluate the differences among mMIC values within the different groups and species, respectively. The Chi-squared test was employed to compare the prevalence of *Malassezia* strains S, SDD, and R to the antifungal drugs tested. A value of *p* ≤ 0.05 was considered statistically significant.

## 3. Results

All quality control MIC values for *C. parapsilosis* determined after 48 h were within the ranges established by the CLSI (0.5–4 for AMB; 1–4 for FLZ; 0.125–0.5 for ITZ; 0.06–0.5 for KTZ; and 0.06–0.25 for POS). For *C. krusei*, the MIC values were 1–4 for AMB; 16–128 for FLZ; 0.25–1 for ITZ and KTZ; and 0.125–1 for POS within 48 h [24]. A good growth of *M. furfur* and *M. sympodialis* was observed in SDB (after 72 h), but for *M. globosa*, we extended the reading time to 5 days because of slow growth (data not shown). 

### 3.1. In Vitro Antifungal Susceptibility Assessment According to the Clinical Origin

The MIC range, mean MIC, MIC50, and MIC90 values obtained for the five antifungal drugs included in this investigation are summarized in Table 1 according to patient group. Overall, FLZ MICs were higher than those observed for other azole drugs and ranged from 32 to >64 mg/L and from 8 to >64 mg/L for isolates from the patient group and control group, respectively. However, POS and ITZ displayed the lowest mMIC, MIC50, and MIC90 values regardless of the clinical origin. In the origin group, the mMICs of FLZ and KTZ were higher in the patient group than in the healthy control group with no significant differences between the groups (*p* = 0.13 and *p* = 0.81, respectively). However, significantly higher mMICs were reported in the patient group with the triazole ITZ (*p* = 0.008). On the other hand, the mMICs of POS and AMB were significantly higher in the control group (*p* = 0.01 and *p* = 0.02, respectively).

### 3.2. In Vitro Antifungal Susceptibility Assessment According to the Species

*Malassezia furfur* displayed the highest mean MICs and MIC90 values to FLZ and ITZ. *Malassezia globosa* showed wide-ranging and higher mean MICs, MIC50, and MIC90 against KTZ, POS, and AMB. *M. sympodialis* seems to be the most susceptible species, especially for FLZ, KTZ, ITZ, and AMB. However, no significant difference in each antifungal activity was noted between the different species (Table 2).

### 3.3. Interpretive Antifungal Susceptibility

Of all the tested isolates, 100% and 98.3% were susceptible to KTZ and POS, respectively, while 43.1%, 29.3%, and 27.6% of strains were found to be resistant to AMB, FLZ, and ITZ, respectively. Regarding the clinical origin, isolates from both origins were susceptible to KTZ and POS (~100%). However, they were less susceptible to FLZ, ITZ, and AMB. The frequency of resistant strains to ITZ and AMB seemed to be higher within healthy skin isolates (44.5% and 66.6% vs. 20% and 30%, respectively). The differences were significant, with *p* = 0.02 and *p* = 0.04 (Table 3).

Regarding the species, 42.1% of *M. furfur* had MICs ≥ 2 mg/L for AMB, 34.2% showed MICs ≥ 64 mg/L for FLZ, and 28.9% had MICs ≥ 1 mg/L for ITZ. Likewise, 33.3% of *M. globosa* had MICs ≥ 2 mg/L for AMB, 33.3% showed MICs ≥ 64 mg/L for FLZ, and 44.5% had MICs ≥ 1 mg/L for ITZ. Of the *M. sympodialis* strains, 45.4% showed MICs ≥ 2 mg/L for AMB. Considering the standards established by the CLSI for *Candida* spp., these isolates would be categorized as resistant (Table 4).

## 4. Discussion

Our findings showed variable in vitro activity for the most used drugs (FLZ, KTZ, ITZ, POS, and AMB) for treating *Malassezia*-related diseases. All the azoles tested showed good antifungal activity except FLZ, which exhibited high mMICs, MIC 50, and MIC90 regardless of the species and the clinical origin of *Malassezia* isolates. The lower antifungal activity of FLZ observed was in agreement with previous studies using similar or different culture media [20,27,28,29,30]. A considerable percentage of FLZ-resistant strains (29.3%) was also revealed in this study. A higher level of FLZ resistance was previously reported using the same methodology [11,21], thus indicating that SDB might be useful for detecting FLZ-resistant isolates. Yet, it is noteworthy that FLZ mMICs from *Malassezia* bloodstream infections (BSI) were higher (demonstrated by wide ranges) than those from human skin [21,27,28]. This finding may imply that BSI isolates developed antifungal resistance as a result of biofilm formation, as previously demonstrated for *M. pachydermatis* [31,32]. Additionally, five *M. furfur* strains showed high FLZ, AMB, and ITZ mMICs (>64, ≥2, and ≥1 mg/L, respectively), six had high AMB and ITZ mMICs (≥2 and ≥1 mg/L, respectively), and two exhibited high FLZ and AMB mMICs (>64 and ≥2mg/L, respectively). This finding is in favor of a cross-resistance phenomenon. Although this event was well documented in *M. pachydermatis* [10,16,27,33,34], this is the second report describing such phenomenon in *M. furfur*. This might explain the poor clinical outcome for some *M. furfur* fungemia [18,21,35,36]. Detailed investigations into resistance mechanisms in human *Malassezia* isolates are needed in order to understand the emergence of new resistant strains. So far, and similar to other fungal species, mutations in *ERG11*, encoding an enzyme in ergosterol biosynthesis and genomic duplications of drug efflux pumps, have been described [37,38,39].

In this study, MIC values obtained for KTZ, POS, and ITZ were considerably lower than those for FLZ. This finding was in line with other studies using different methodologies [19,21,27,40]. However, KTZ and POS seem to be the most active molecules against *Malassezia* spp. In fact, 100% and 98.3% of strains were found to be susceptible, respectively. This higher activity of KTZ was reported by many authors in different studies [19,27,41,42,43]. Ketoconazole is likely the most extensively studied azole compound with good in vitro activity, making it an excellent choice for the topical treatment of *Malassezia*-related conditions [27]. However, this azole is no longer recommended as first-line treatment because of its toxicity [19]. Thus, POS and ITZ could be effective alternative options, as shown in previous studies using the CLSI protocol [20,27,28]. Nevertheless, a sizable percentage of ITZ-resistant isolates (27.6%) was reported in this study. Thus, POS might be used as an optimal antifungal agent for the management of *pityriasis versicolor* and other skin diseases. 

As for AMB, the M27-A3 document states that *Candida* species with MICs ≥ 2 mg/L are likely resistant. Accordingly, we recorded high mean AMB mMICs, MIC50, and MIC90 and wide MIC ranges, particularly within *M. furfur* and *M. globosa*. In addition, 43.1% of the isolates showed MIC > 1 mg/L. The low in vitro efficacy of AMB registered herein is concordant with other previous findings [19,21,28,30,41]. Since standard methods for determining the antifungal susceptibility of *Malassezia* spp. have not been validated by a consensus procedure, these data need to be confirmed by assessing the clinical outcome of this drug.

According to the clinical origin, significantly higher mMICs with the triazole ITZ were reported within the patient group. On the other hand, POS and AMB MIC data were surprisingly higher from healthy skin than lesional skin. In addition, the percentages of resistant strains to ITZ and AMB found within controls were significantly higher than those reported within PV patients (*p* = 0.02 and 0.04, respectively), which was not expected. To our knowledge, studies evaluating the in vitro susceptibility of *Malassezia* from healthy and lesioned skin are scant, making strong statements about our findings difficult. In fact, the sole study comparing the in vitro susceptibility of *M. pachydermatis* isolated from healthy and lesioned skins was carried out on a canine population. That study revealed higher MIC50 for KTZ and POS in skin lesions than in healthy skin [44]. In addition, the small number of *Malassezia* isolated from healthy controls (i.e., *n* = 18) used in the current study and the lack of prior investigations on the subject prevent us from drawing an evidence-based conclusion about the relationship between the occurrence of lesions and the antifungal profile. This topic requires further investigation. Studying the antifungal susceptibility profiles of *Malassezia* isolates coming from skin and bloodstream infections might also be interesting in order to assess the impact of the clinical origin on the antifungal profiles of these yeast species.

Regarding the species, the highest MICs with the widest ranges for all drugs tested, especially FLZ, KTZ, and ITZ, were for *M. furfur*. In addition, 42.1%, 34.2%, and 28.9% of strains would be categorized as resistant for AMB, FLZ, and KTZ, respectively. Similar results were obtained by others using modified microdilution methods [19,27]. In contrast, POS was the most active drug against this species, with low MICs and limited variation in susceptibility among different isolates. 

*Malassezia globosa* also exhibited high MICs, particularly to FLZ, KTZ, POS, and AMB. In contrast, all isolates of *M. globosa* tested were susceptible in vitro to KTZ (MIC ≤ 8 mg/L). These findings are comparable with those of Rojas et al. [19]. According to previous research [19,27,41,42,43,45], *M. globosa* is one of the least susceptible species to antifungal drugs. Indeed, in the current study, two isolates of *M. globosa* exhibited high MICs for FLZ, ITZ, and AMB (64 mg/L, ≥1mg/L, and ≥4 mg/L, respectively), thus suggesting a possible cross-resistance in *M. globosa* as well. Rojas et al. previously reported the presence of *M. globosa* isolates showing high MICs for FLZ, MCZ, and AMB [19]. In addition, Miranda et al. described a similar case of one isolate showing low susceptibility to ITZ, VCZ, and FLZ [43].

Comparably, *M. sympodialis* had almost the lowest MICs and the lowest intraspecies variation. Ketoconazole, ITZ, and AMB were the most active drugs against this species. This observation is in line with previous studies considering *M. sympodialis* one of the most susceptible species to all antifungals [19,27,45].

The differences observed between *Malassezia furfur*, *M. sympodialis*, and *M. globosa* emphasize the importance of accurately identifying and evaluating the antifungal susceptibility of the three major pathogenic *Malassezia* species in human diseases [19]. Finally, in the absence of clinical break points, experts recently agreed that epidemiological cut-off values (ECVs) could be beneficial for differentiating susceptible (i.e., wild-type (WT)) and resistant (i.e., non-WT) isolates, and are thus useful for monitoring the emergence of isolates with decreased susceptibilities [20,46,47,48]. However, “tentative” azole ECVs were solely established for *M. pachydermatis* and *M. furfur* isolated from human bloodstream infections [20]. Hence, additional and multicenter studies are needed to set epidemiological and clinical susceptibility breakpoints for this genus.

In conclusion, regardless of the species, *Malassezia* yeasts are susceptible to POS, ITZ, and KTZ but less so to FLZ and AMB. Although FLZ, ITZ, and AMB are among the most commonly used antifungal agents worldwide, our results showed higher MICs and higher percentages of ‘‘resistant” strains than for other molecules. This study may constitute an alarm for the scientific community regarding rising resistance in clinical *Malassezia* isolates. Nonetheless, the interspecies variability of the MIC distribution highlights the importance of defining each species’ susceptibility profile in order to obtain reliable information for implementing an effective treatment regimen against *Malassezia*-related diseases [20].

## Figures and Tables

**Table 1 jof-08-00959-t001:** Fluconazole (FLZ), ketoconazole (KTC), itraconazole (ITZ), posaconazole (POS), and amphotericin B (AMB) minimum inhibitory concentration (MIC, mg/L) data, standard deviation (SD), MIC range, MIC50, and MIC90 of *Malassezia* spp. according to the clinical origin.

*Malassezia* Species Origin	Fluconazole				Ketoconazole				Itraconazole				Posaconazole				Amphotericin B			
	Range	Mean MIC (SD)	MIC50	MIC90	Range	Mean MIC (SD)	MIC50	MIC90	Range	Mean MIC (SD)	MIC50	MIC90	Range	Mean MIC (SD)	MIC50	MIC90	Range	Mean MIC (SD)	MIC50	MIC90
Group 1 (40 isolates)	32->64	40.1 (16.2)	32	64	0.125–8	1.72 (2.4)	0.5	4	0.06–8	0.95 * (2.16)	0.125	1	0.06–2	0.22 * (0.34)	0.125	0.25	0.25–8	1.79 * (2.08)	1	4
Group 2 (18 isolates)	4->64	35.6 (21.2)	32	64	0.06–4	0.91 (0.89)	0.5	1	0.06–2	0.61 * (0.37)	0.5	1	0.125–1	0.35 * (0.26)	0.25	0.5	0.5–4	2.42 * (1.39)	2	4
Total	4->64	38 (17.7)	32	64	0.06–8	1.60 (2.3)	0.5	4	0.06–8	0.90 (1.8)	0.25	2	0.06–2	0.26 (0.3)	0.125	0.5	0.25–8	2.02 (1.9)	1	4

* The difference is statistically significant with *p* value < 0.05.

**Table 2 jof-08-00959-t002:** Fluconazole (FLZ), ketoconazole (KTC), itraconazole (ITZ), posaconazole (POS), and amphotericin B (AMB) minimum inhibitory concentration (MIC, mg/L) data, standard deviation (SD), MIC range, MIC50, and MIC90 of *Malassezia* isolates according to the species.

Malassezia Species	Fluconazole				Ketoconazole				Itraconazole				Posaconazole				Amphotericin B			
	Range	Mean MIC (SD)	MIC50	MIC90	Range	Mean MIC (SD)	MIC50	MIC90	Range	Mean MIC (SD)	MIC50	MIC90	Range	Mean MIC (SD)	MIC50	MIC90	Range	Mean MIC (SD)	MIC50	MIC90
** *M. furfur* **	4->64	39.9 (18.1)	32	64	0.125–8	1.91 (2.53)	0.5	4	0.06–8	1.03 (2.19)	0.125	2	0.06–1	0.22 (0.2)	0.125	0.5	1–8	2.03 (1.93)	2	4
** *M. sympodialis* **	8–64	31.2 (14.34)	32	32	0.06–1	0.43 (0.25)	0.5	0.5	0.06–2	0.62 (0.75)	0.25	2	0.06–1	0.32 (0.3)	0.125	0.5	0.25–4	1.75 (1.35)	1	4
** *M. globosa* **	8->64	37. 7(19.9)	32	64	0.5–8	1.63 (2.36)	0.5	2	0.06–2	0.6 (0.7)	0.25	1	0.06–2	0.4 (0.66)	0.125	0.5	0.5–8	2.3 (2.6)	1	4

**Table 3 jof-08-00959-t003:** Interpretive antifungal susceptibility classification of *Malassezia* isolates according to the clinical origin into susceptible (S), susceptible dose-dependent (SDD), and resistant (R). Number of isolates (and percentage) by category.

*Malassezia* Species Origin	Fluconazole			Ketoconazole			Itraconazole			Posaconazole			Amphotericin B		
	S	SDD	R	S	SDD	R	S	SDD	R	S	SDD	R	S	SDD	R
Group 1 (40 isolates)	0 (0)	28 (70)	12 (30)	40 (100)	-	0 (0)	26 (65)	6 (15)	8 (20)	39 (97.5)	1 (0.02)	0 (0)	28 (70)	-	12 (30)
Group 2 (18 isolates)	5 (27.8)	8 (44.4)	5 (27.8)	18 (100)	-	0 (0)	4 (22.2)	6 (33.3)	8 (44.5) *	18 (100)	0 (0)	0 (0)	6 (33.3)	-	12 (66.6) *
Total	5 (8.6)	36 (62.1)	17 (29.3)	58 (100)	-	0 (0)	30 (51.7)	12 (20.7)	16 (27.6)	57 (98.3)	1 (1.7)	0 (0)	34 (58.6)	-	24 (41.4)

* The difference is statistically significant with *p* value < 0.05.

**Table 4 jof-08-00959-t004:** Interpretive antifungal susceptibility classification of *Malassezia* species according to the species into susceptible (S), susceptible dose-dependent (SDD), and resistant (R): number of isolates (and percentage) by category.

	Species	*M. furfur* (*n* = 38)			*M. sympodialis* (*n* = 11)			*M. globosa* (*n* = 9)		
Antifungal Agent		S	SDD	R	S	SDD	R	S	SDD	R
Fluconazole		1 (2.6)	24 (36.2)	13 (34.2)	2 (18.2)	8 (72.7)	1 (9.1)	2 (22.2)	4 (44.5)	3 (33.3)
Ketoconazole		38 (100)	0 (0)	0 (0)	11 (100)	0 (0)	0 (0)	9 (100)	0 (0)	0 (0)
Itraconazole		22 (57.9)	5 (13.2)	11 (28.9)	4 (36.4)	5 (45.4)	2 (18.2)	4 (44.5)	2 (22.2)	3 (33.3)
Posaconazole		38 (100)	0 (0)	0 (0)	11 (100)	0 (0)	0 (0)	8 (88.9)	1 (11.1)	0 (0)
Amphotericin B		22 (57.9)	0 (0)	16 (42.1)	6 (54.6)	0 (0)	5 (45.4)	5 (55.5)	0 (0)	4 (44.5)

## Data Availability

All original data can be obtained from the authors.

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
