# Peer review of "In Vitro Assessment of Azole and Amphotericin B Susceptibilities of Malassezia spp. Isolated from Healthy and Lesioned Skin"

_jof, 2022, doi:10.3390/jof8090959_

Round 1
Reviewer 1 Report (Previous Reviewer 3)
I agree to be accepted in the present form
Author Response
We thank the referee for evaluating the manuscript
Reviewer 2 Report (New Reviewer)
Comments and Suggestions for Authors
1. There is a lot of work that investigates this field. What is the novelty of this work? Please highlight the novelty of this work.
2. The English used should be revised by a native speaker, since many sentences cannot be properly understood and there are a lot of grammar mistakes.
3. Need to include future directions/ideas in the discussion section.
-L28, 43.1% shall be 41.4%
Introduction
-L42, M. furfur shall be italicized
MM section
-L82~87 shall be deleted.
-L116~117, the sentence is not clear or understandable. “Data obtained were reported as MIC ranges, mean MICs (mMICs), MIC at which 50% (MIC50) and 90% (MIC90) of the isolates were inhibited.”
Result:
-L133~138, the sentence is not clear or understandable. The unit shall be listed.
-L163, “MIC90 ± MIC50 values”. Please check it.
-L177, 43.1%, Please check it. It shall be 41.4%.
-L185, 44.5%, Please check it. It shall be 33.3%
There is no words to describe about M. sympodialis in section 3.3.
Discussion
-L247, Besides, 43.1% of the isolates showed MIC > 1 mg/L. Please check it.

Author Response
1. There is a lot of work that investigates this field. What is the novelty of this work? Please highlight the novelty of this work.
Actually it is stated and end of introduction that data is scarce where healthy and diseased people are compared. We feal it is important in the light of antimicrobial resistance that from this work it is clear that also resistant isolates are isolated from healthy individuals.
3. Need to include future directions/ideas in the discussion section.
Studying antifungal susceptibility profiles of Malassezia isolates coming from skin and bloodstream infections might also be interesting in order to assess the impact of the clinical origin on the antifungal profiles of these yeast species.
We have rephrased this and added this also into the discussion.
-L133~138, the sentence is not clear or understandable. The unit shall be listed.
We have adapted the sentences and now made two sentences, one for each species. We have now changed the MIC values in the median values that we obtained for all the experiments. For every MIC experiment, the control species where each time also incorporated. There is always a small variability and therefore we now added the median values.
-L163, “MIC90 ± MIC50 values”. Please check it.
This expression has been replaced and the text was made more clear.
-L177, 43.1%, Please check it. It shall be 41.4%.
43.1% is the right value. It has been corrected accordingly in the abstract.
-L185, 44.5%, Please check it. It shall be 33.3%.
The authors agree. The value has been changed accordingly in the text.
There is no words to describe about M. sympodialis in section 3.3.
The authors agree. A sentence about M. sympodialis was added in the section 3.3.
Discussion
-L247, Besides, 43.1% of the isolates showed MIC > 1 mg/L. Please check it.
It is correct. Indeed, the right value is 43.1%.
Reviewer 3 Report (New Reviewer)
The manuscript presents the in vitro assessment of the antifungal susceptibility profile of 58 Malassezia strains, isolated from patients with Pityriasis versicolor or healthy individuals. In my opinion, the manuscript covers an interesting topic, the research is well conducted and the paper is well written. The paper can be accepted in its current form.
Author Response
We thank the referee for evaluating the manuscript
Round 2
Reviewer 2 Report (New Reviewer)
I have no further suggestion.
This manuscript is a resubmission of an earlier submission. The following is a list of the peer review reports and author responses from that submission.
Round 1
Reviewer 1 Report
Dear Editors
This manuscript has main concern as follows:
1. The subject is not very important in medical mycology.
2. Some parts of introduction and discussion sections are out of topic (see attached file).
3. The selected region for identification of Malassezia spp. is not appropriate.
4. The identification method is described very briefly and even the sequence of primers, PCR conditions are not mentioned.
5. Authors used the old version of CLSI (M27-A3, 2008), whereas the updated version (M27 4th edition, 2018) has been released.

Author Response
Referee #1
- The subject is not very important in medical mycology.
Actually, we do not agree with the reviewer regarding this point. Indeed, it is important to highlight that the epidemiology of fungal diseases has greatly changed over the past few decades. In particular, Malassezia species have recently emerged as pathogens associated with a wide range of dermatological diseases and systemic infections (Rhimi et al. 2020). Besides, since standardized methods for in vitro evaluation of antifungal susceptibility have not yet been established for Malassezia spp., related infections are always treated empirically. As a result, a high rate of recurrence and a decreased antifungal susceptibility have appeared (Shah et al. 2013; Pedrosa et al. 2014). In fact, reports on resistance phenomena are rising steadily from all over the world (Leong et al. 2021). Thus, because reduced in vitro antifungal susceptibility may be an indicator of clinical failure, there is a need to profile the susceptibility of Malassezia isolates derived from individuals with healthy or disease backgrounds. To sum up, assessment of in vitro susceptibility profiles of Malassezia yeasts is an extremely important subject that can help clinicians guide an accurate and evidence-based treatment, and consequently decrease the threatof invasive infections.
- Some parts of introduction and discussion sections are out of topic (see attached file).
We do not agree with the referee that the first part of the introduction is out of scope. As Journal of Fungi is a broad journal, not everyone that will read this is specialist in this topic.
- The selected region for identification of Malassezia spp. is not appropriate.
During this study, we used two identification approaches: the phenotypic characterization (the conventional method) and the molecular approach by multiplex PCR of the ITS region. It is true that the ITS region is a highly variable region within fungal genomes. However, it has been successfully used for the identification of Malassezia to the species level (Gupta et al.2000; Affes et al. 2009; Cafarchia et al. 2011) and has been proven to detect intra-specific polymorphism. Besides, in the current study, the molecular characterization was confirmed by PCR sequencing (Gaitanis et al.2006) (as the sequenced DNA samples have served as positive controls for the multiplex PCR) as well as PCR-RFLP (Chebil et al.2022).
- The identification method is described very briefly and even the sequence of primers, PCR conditions are not mentioned.
The authors agree. More details related to the protocol were added in the Materials and Methods section.
- Authors used the old version of CLSI (M27-A3, 2008), whereas the updated version (M27 4th edition, 2018) has been released.
The authors agree, however since there was no update regarding Candida breakpoints in the new version of 2018, we kept the old one as a reference.
Reviewer 2 Report
The authors constructed in vitro antifungal susceptibility testing system for Malassezia species and got some findings about drug resistance of this fungi.
My comments are shown as the below.
1. The authors may consider reorganizing the appearance of Table 1 and Table 2 for easier understanding. The author also needs to clarify in the title of the table which data were statistically significantly different and add in the title if there is any symbol used for marking the significant difference. There were several “bold” format used in all the table but the authors did not mention the meaning in the manuscript.
2. If the authors divided the isolates into two groups, please also show the antifungal susceptibility results of each group in Table 3 and interpret the difference between the isolates from patients and the ones from healthy controls.
3. Since the author already mentioned in the discussion part, ERG11 is important to the study in the resistance mechanism of Malassezia and other pathogenic fungi, they should perform the sequence analysis of ERG11 gene in all the isolates.
4. Because Candida spp. criteria were used for data interpretation in this study, the author should include some Candida strains in each MIC test as the control strains.
5. Line 164, could the author explain the part “28.9% had MICs ≥ 16 µg/mL for ITZ”, since the author wrote in Table 1 and Table 2 that the ITZ drug range used was 0.06-8 µg/mL? Also, similar comment in Line 166 where “44.5% had MICs ≥ 16 µg/mL”.
6. Line 32, Line 91-92, re-format the words color to be similar to the whole manuscript.
7. Line 137, 141, 164, 165, and 234. Replace “FCZ” with “FLZ”. There were inconsistencies in the fluconazole abbreviation.
8. Line 167, replace “Candida” with ‘Candida spp.”
9. Line 189-192, the authors mentioned this is the first report describing the cross-resistance phenomenon in M. furfur. However, from reference (24) Iata et al., results mentioned about this phenomenon too in M.furfur. Does the authors mean the first phenomenon on the cross-resistance for FLZ, AMB, and ITZ only?
10. In the discussion part, the authors did not discuss whether each Malassezia species have any relation with its clinical origin. Was the M. furfur/M. globosa/M. sympodialis isolated from the lesion skin more resistant or more susceptible to azole drugs and AMB compared to M. furfur/M. globosa/M. sympodialis isolated from the healthy skin? The author may elaborate more whether or not the skin lesions were directly/indirectly affecting the antifungal profile for each Malassezia strain used in this study.
11. Line 238-239. Could the authors explain or rephrase the sentences? Since the KTZ were mentioned twice which the sentences have the opposite meaning.
12. Line 260-261, could the authors rephrase the sentence for clarity.
“Hence, the importance of additional and multicenter studies ‘is needed’ to set epidemiological and clinical susceptibility breakpoints for this genus.” or etc.
Author Response
Referee#2
- The authors may consider reorganizing the appearance of Table 1 and Table 2 for easier understanding. The author also needs to clarify in the title of the table which data were statistically significantly different and add in the title if there is any symbol used for marking the significant difference. There were several “bold” format used in all the table but the authors did not mention the meaning in the manuscript.
The tables’appearance has been changed as recommended by the reviewer and the authors propose the new version of tables (landscape format). The bold format has been deleted. Regarding the statistically significant difference, it was marked as a footnote but deleted by mistake throughout the submission process, so we reincluded it in the footnote.
- If the authors divided the isolates into two groups, please also show the antifungal susceptibility results of each group in Table 3 and interpret the difference between the isolates
Done.
- Since the author already mentioned in the discussion part, ERG11 is important to the study in the resistance mechanism of Malassezia and other pathogenic fungi, they should perform the sequence analysis of ERG11 gene in all the isolates
We agree with the reviewer that ERG11 can be an interesting target to study the resistance mechanisms of Malassezia. This investigation is already planned as a research project in the future, where also the sterol pattern will be evaluated.
- Because Candida spp. criteria were used for data interpretation in this study, the author should include some Candidastrains in each MIC test as the control strains.
In our humble opinion, there is no interest in including other Candida strains in each MIC test, since all quality control MIC values (Candida parapsilosis and Candida krusei) determined were within the ranges established by the CLSI guidelines.
- Line 164, could the author explain the part “28.9% had MICs ≥ 16 µg/mL for ITZ”, since the author wrote in Table 1 and Table 2 that the ITZ drug range used was 0.06-8 µg/mL?  Also, similar comment in Line 166 where “44.5% had MICs ≥ 16 µg/mL”.
The authors agree. Indeed, according to CLSI criteria for Candida, resistance to ITZ is considered when MICs ≥ 1 µg/mL and not 16 µg/mL. The number 6 has been added by mistake. The value hasbeen corrected in the lines188 and 190.
- Line 32, Line 91-92, re-format the words color to be similar to the whole manuscript.
The words color has been adjusted as recommended by the reviewer.
- Line 137, 141, 164, 165, and 234. Replace “FCZ” with “FLZ”. There were inconsistencies in the fluconazole abbreviation.
The authors agree. The abbreviation has been standardized.
- Line 167, replace “Candida” with ‘Candida spp.”
Done.
- Line 189-192, the authors mentioned this is the first report describing the cross-resistance phenomenon in M. furfur. However, from reference (24) Iatta et al., results mentioned about this phenomenon too in M.furfur. Does the authors mean the first phenomenon on the cross-resistance for FLZ, AMB, and ITZ only?
Yes. Indeed, Iatta et al described the occurrence of a possible cross-resistance phenomenon in M. furfur strains within FLC, ITZ, POS and VRZ, but not for AMB.
- In the discussion part, the authors did not discuss whether each Malassezia species have any relation with its clinical origin. Was the M. furfur/M. globosa/M. sympodialis isolated from the lesion skin more resistant or more susceptible to azole drugs and AMB compared to M. furfur/M. globosa/M. sympodialis isolated from the healthy skin? The author may elaborate more whether or not the skin lesions were directly/indirectly affecting the antifungal profile for each Malassezia strain used in this study.
The authors agree with the reviewer regarding the first part of the comment. This has been accordingly added to the discussion section. Regarding whether we may elaborate more or not if the skin lesions were directly or indirectly affecting the antifungal profile for each Malassezia strain used in this study, this seems controversial as the activity of the antifungals used in this study was variable according to the clinical origin. For instance, this study revealed higher MIC50 for KTZ and POS and higher mMICs for ITZ in skin lesions than in healthy skin, POS and AMB MIC data were higher from healthy skin than lesional skin. Besides, as mentioned in the discussion, the small number of Malassezia isolated from healthy controls (i.e., n=18), prevents us to draw an evidence-based conclusion and requires further investigation.
- Line 238-239. Could the authors explain or rephrase the sentences? Since the KTZ were mentioned twice which the sentences have the opposite meaning.
The authors agree. The first term KTZ has been put by mistake instead of ITZ, so it has been removed and replaced by ITZ.
- Line 260-261, could the authors rephrase the sentence for clarity.
“Hence, the importance of additional and multicenter studies ‘is needed’ to set epidemiological and clinical susceptibility breakpoints for this genus.” or etc.
The authors propose this sentence ''Hence, additional and multicenter studies are needed in order to set epidemiological and clinical susceptibility breakpoints for this genus"
Reviewer 3 Report
The work by Chebil et al., is interesting and the data concerning Malassezia spp. susceptibility very useful to future treatment of Malassezia infections.
My suggestion is to be accepted after minor modifications
1. Antibiotic concentration presentation must be kept constant and not change from μg/ml to mgr/L
2. Indicators of MIC percent should be presented as subscripts
3. Data for species molecular identification must be presented in the results
4. Details for samples collection must be presented in the manuscript
5. MIC were calculated graphically or by a computer program? please give details
6. References must be checked again: follow instructions, add pages, bold volumes, italic journal etc..
Author Response
Referee #3
- Antibiotic concentration presentation must be kept constant and not change from μg/ml to mgr/L
The authors agree. The unit(mg/L) has been standardized along the manuscript.
- Indicators of MIC percent should be presented as subscripts
Done.
- Data for species molecular identification must be presented in the results
It is not clear what the referee means. Do you mean agarose gels for the different PCR products? The different nucleotide sequences?
- Details for samples collection must be presented in the manuscript
The samples were randomly selected as well as from clinically diagnosed patients.
- MIC were calculated graphically or by a computer program? please give details
MIC values were determined visually as already mentioned in the text, line 106.
- References must be checked again: follow instructions, add pages, bold volumes, italic journal etc..
Done.
Round 2
Reviewer 1 Report
Dear Editor
My opinion on this manuscript has not changed.
I was not satisfied with the authors' answers.
Kind Regards,
Rasoul Mohammadi
Reviewer 2 Report
- The authors may consider reorganizing the appearance of Table 1 and Table 2 for easier understanding. The author also needs to clarify in the title of the table which data were statistically significantly different and add in the title if there is any symbol used for marking the significant difference. There were several “bold” format used in all the table but the authors did not mention the meaning in the manuscript.
The tables’appearance has been changed as recommended by the reviewer and the authors propose the new version of tables (landscape format). The bold format has been deleted. Regarding the statistically significant difference, it was marked as a footnote but deleted by mistake throughout the submission process, so we reincluded it in the footnote.
>>The authors’ answer is acceptable for me. Although after reading carefully again in Table 1, the “Total” for KTZ was written “8” and ITZ was written “1”, which were beyond the MIC90. Could the authors confirm that those data written correctly? Or could the author explain this in the manuscript?
- If the authors divided the isolates into two groups, please also show the antifungal susceptibility results of each group in Table 3 and interpret the difference between the isolates
Done.
>>The authors’ answer is acceptable for me. However, in Table 3, the value inside the bracket should be mentioned as percentage. This is to prevent misunderstanding that this value is not standard deviation (SD) as what was written for Table 1 and Table 2.
- Since the author already mentioned in the discussion part, ERG11 is important to the study in the resistance mechanism of Malassezia and other pathogenic fungi, they should perform the sequence analysis of ERG11 gene in all the isolates
We agree with the reviewer that ERG11 can be an interesting target to study the resistance mechanisms of Malassezia. This investigation is already planned as a research project in the future, where also the sterol pattern will be evaluated.
>> I don’t consider that the current data is enough to publish a paper. The authors need to add the data of sequence analysis of ERG11 gene in all the isolates and sterol evaluation. They put the data in the manuscript.
- Because Candida spp. criteria were used for data interpretation in this study, the author should include some Candidastrains in each MIC test as the control strains.
In our humble opinion, there is no interest in including other Candida strains in each MIC test, since all quality control MIC values (Candida parapsilosis and Candida krusei) determined were within the ranges established by the CLSI guidelines.
>>The authors may show the control MIC values of C. parapsilosis and C. krusei in supplementary table.
- Line 164, could the author explain the part “28.9% had MICs ≥ 16 µg/mL for ITZ”, since the author wrote in Table 1 and Table 2 that the ITZ drug range used was 0.06-8 µg/mL?  Also, similar comment in Line 166 where “44.5% had MICs ≥ 16 µg/mL”.
The authors agree. Indeed, according to CLSI criteria for Candida, resistance to ITZ is considered when MICs ≥ 1 µg/mL and not 16 µg/mL. The number 6 has been added by mistake. The value hasbeen corrected in the lines188 and 190.
>>The author’s answer to comment no.5 is not properly corrected in the manuscript. The edited manuscript is still written “16 mg/L” instead of “1 mg/L”, shown in line 171 and line 173. On the other hand, there were no corrections and correlations found in line 188 and line 190 as the author answered.
- In the discussion part, the authors did not discuss whether each Malassezia species have any relation with its clinical origin. Was the M. furfur/M. globosa/M. sympodialis isolated from the lesion skin more resistant or more susceptible to azole drugs and AMB compared to M. furfur/M. globosa/M. sympodialis isolated from the healthy skin? The author may elaborate more whether or not the skin lesions were directly/indirectly affecting the antifungal profile for each Malassezia strain used in this study.
The authors agree with the reviewer regarding the first part of the comment. This has been accordingly added to the discussion section. Regarding whether we may elaborate more or not if the skin lesions were directly or indirectly affecting the antifungal profile for each Malassezia strain used in this study, this seems controversial as the activity of the antifungals used in this study was variable according to the clinical origin. For instance, this study revealed higher MIC50 for KTZ and POS and higher mMICs for ITZ in skin lesions than in healthy skin, POS and AMB MIC data were higher from healthy skin than lesional skin. Besides, as mentioned in the discussion, the small number of Malassezia isolated from healthy controls (i.e., n=18), prevents us to draw an evidence-based conclusion and requires further investigation.
>>I still consider that the discussion is too brief in line 230-241, although the authors mentioned that interpretation of this finding is difficult. Could the authors discuss the tendency that the results showed in this study, if it is too controversial to state the direct/indirect effect? And also discuss why there were unexpected result in line 232-234.
- Line 238-239. Could the authors explain or rephrase the sentences? Since the KTZ were mentioned twice which the sentences have the opposite meaning.
The authors agree. The first term KTZ has been put by mistake instead of ITZ, so it has been removed and replaced by ITZ.
>>The author’s answer is not properly corrected in the manuscript. The edited manuscript is still written “KTZ” twice in lines 248-249 instead of “ITZ” correction as the authors’ answered.